# Compacting, Picking and Growing for Unforgetting Continual Learning

**Steven C. Y. Hung, Cheng-Hao Tu, Cheng-En Wu, Chien-Hung Chen,
Yi-Ming Chan, and Chu-Song Chen**
Institute of Information Science, Academia Sinica, Taipei, Taiwan
MOST Joint Research Center for AI Technology and All Vista Healthcare
{brent12052003, andytu455176}@gmail.com,
{chengen, redsword26, yiming, song}@iis.sinica.edu.tw

## Abstract

Continual lifelong learning is essential to many applications. In this paper, we
propose a simple but effective approach to continual deep learning. Our approach
leverages the principles of deep model compression, critical weights selection,
and progressive networks expansion. By enforcing their integration in an iterative
manner, we introduce an incremental learning method that is scalable to the number
of sequential tasks in a continual learning process. Our approach is easy to imple-
ment and owns several favorable characteristics. First, it can avoid forgetting (i.e.,
learn new tasks while remembering all previous tasks). Second, it allows model
expansion but can maintain the model compactness when handling sequential tasks.
Besides, through our compaction and selection/expansion mechanism, we show
that the knowledge accumulated through learning previous tasks is helpful to build a
better model for the new tasks compared to training the models independently with
tasks. Experimental results show that our approach can incrementally learn a deep
model tackling multiple tasks without forgetting, while the model compactness is
maintained with the performance more satisfiable than individual task training.

## 1   Introduction

Continual lifelong learning [42, 28] has received much attention in recent deep learning studies. In
this research track, we hope to learn a model capable of handling unknown sequential tasks while
keeping the performance of the model on previously learned tasks. In continual lifelong learning, the
training data of previous tasks are assumed non-available for the newly coming tasks. Although the
model learned can be used as a pre-trained model, fine-tuning a model for the new task will force the
model parameters to fit new data, which causes *catastrophic forgetting* [24, 31] on previous tasks.

To lessen the effect of catastrophic forgetting, techniques leveraging on regularization of gradients
or weights during training have been studied [14, 49, 19, 35]. In Kirkpatrick et al. [14] and Zenke
et al. [49], the proposed algorithms regularize the network weights and hope to search a common
convergence for the current and previous tasks. Schwarz et al. [40] introduce a network-distillation
method for regularization, which imposes constraints on the neural weights adapted from the teacher to
the student network and applies the elastic-weight-consolidation (EWC) [14] for incremental training.
The regularization-based approaches reduce the affection of catastrophic forgetting. However, as the
training data of previous tasks are missing during learning and the network capacity is fixed (and
limited), the regularization approaches often forget the learned skills gradually. Earlier tasks tend to
be forgotten more catastrophically in general. Hence, they would not be a favorable choice when the
number of sequential tasks is unlimited.

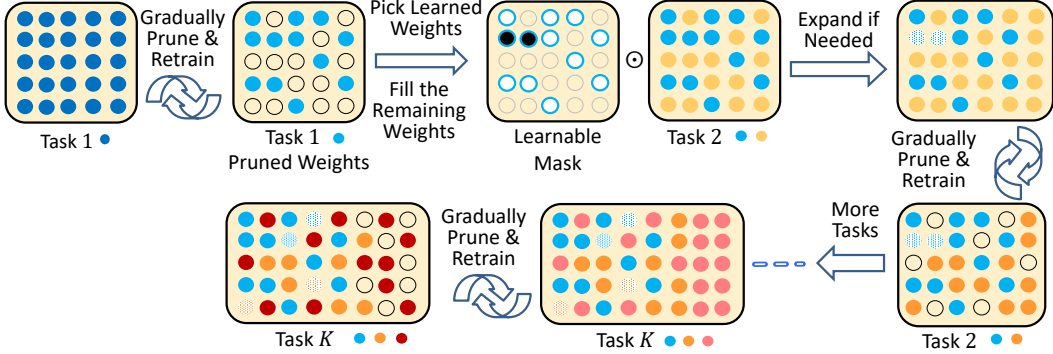

Figure 1: Compacting, Picking, and Growing (CPG) continual learning. Given a well-trained model, gradual pruning is applied to compact the model to release redundant weights. The compact model weights are kept to avoid forgetting. Then a learnable binary weight-picking mask is trained along with previously released space for new tasks to effectively reuse the knowledge of previous tasks. The model can be expanded for new tasks if it does not meet the performance goal. Best viewed in color.

To address the data-missing issue (i.e., lacking of the training data of old tasks), data-preserving and memory-replay techniques have been introduced. Data-preserving approaches (such as [32, 3, 11, 34]) are designed to directly save important data or latent codes as an efficient form, while memory-replay approaches [41, 46, 13, 45, 27] introduce additional memory models such as GANs for keeping data information or distribution in an indirect way. The memory models have the ability to replay previous data. Based on past data information, we can then train a model such that the performance can be recovered to a considerable extent for the old tasks. However, a general issue of memory-replay approaches is that they require explicit re-training using old information accumulated, which leads to either large working memory or compromise between the information memorized and forgetting.

This paper introduces an approach for learning sustainable but compact deep models, which can handle an unlimited number of sequential tasks while avoiding forgetting. As a limited architecture cannot ensure to remember the skills incrementally learned from unlimited tasks, our method allows growing the architecture to some extent. However, we also remove the model redundancy during continual learning, and thus can increasingly compact multiple tasks with very limited model expansion.

Besides, pre-training or gradually fine-tuning the models from a starting task only incorporates prior knowledge at initialization; hence, the knowledge base is getting diminished with the past tasks. As humans have the ability to continually acquire, fine-tune and transfer knowledge and skills throughout their lifespan [28], in lifelong learning, we would hope that the experience accumulated from previous tasks is helpful to learn a new task. As the model increasingly learned by using our method serves as a compact, un-forgetting base, it generally yields a better model for the subsequent tasks than training the tasks independently. Experimental results reveal that our lifelong learning method can leverage the knowledge accumulated from the past to enhance the performance of new tasks.

**Motivation of Our Method Design:** Our method is designed by combining the ideas of deep model compression via weights pruning (Compacting), critical weights selection (Picking), and ProgressiveNet extension (Growing). We refer it to as CPG, whose rationals are given below.

As stated above, although the regularization or memory-replay approaches lessen the effect of forgetting, they often do not guarantee to preserve the performance for previous tasks. To exactly avoid forgetting, a promising way is to keep the old-task weights already learned [38, 47] and enlarge the network by adding nodes or weights for training new tasks. In ProgressiveNet [38], to ease the training of new tasks, the old-task weights are shared with the new ones but remain fixed, where only the new weights are adapted for the new task. As the old-task weights are kept, it ensures the performance of learned tasks. However, as the complexity of the model architecture is proportional to the number of tasks, it yields a highly redundant structure for keeping multiple models.

Motivated by ProgressiveNet, we design a method allowing the sustainability of architecture too. To avoid constructing a complex and huge structure like ProgressiveNet, we perform model compression for the current task every time so that a condensed model is established for the old tasks. According to deep-net compression [10], much redundancy is contained in a neural network and removing the

redundant weights does not affect the network performance. Our approach exploits this property, which compresses the current task by deleting neglectable weights. This yields a compressing-and-growing loop for a sequence of tasks. Following the idea of ProgressiveNet, the weights preserved for the old tasks are set as invariant to avoid forgetting in our approach. However, unlike ProgressiveNet where the architecture is always grown for a new task, as the weights deleted from the current task can be released for use for the new tasks, we do not have to grow the architecture every time but can employ the weights released previously for learning the next task. Therefore, in the **growing** step of our CPG approach, two possible choices are provided. The first is to use the previously released weights for the new task. If the performance goal is not fulfilled yet when all the released weights are used, we then proceed to the second choice where the architecture is expanded and both the released and expanded weights are used for the new-task training.

Another distinction of our approach is the "picking" step. The idea is motivated below. In ProgressiveNet, the old-tasks weights preserved are all co-used (yet remain fixed) for learning the new tasks. However, as the number of tasks is increased, the amount of old-task weights is getting larger too. When all of them are co-used with the weights newly added in the growing step, the old weights (that are fixed) act like inertia since only the fewer new weights are allowed to be adapted, which tends to slow down the learning process and make the solution found immature in our experience. To address this issue, we do not employ all the old-task weights but picking only some critical ones from them via a differentiable mask. In the **picking** step of our CPG approach, the old weights' picking-mask and the new weights added in the growing step are both adapted to learn an initial model for the new task. Then, likewise, the initial model obtained is compressed and preserved for the new task as well.

To compress the weights for a task, a main difficulty is the lacking of prior knowledge to determine the pruning ratio. To solve this problem, in the **compacting** step of our CPG approach, we employ the gradual pruning procedure [51] that prunes a small portion of weights and retrains the remaining weights to restore the performance iteratively. The procedure stops when meeting a pre-defined accuracy goal. Note that only the newly added weights (from the released and/or expanded ones in the growing step) are allowed to be pruned, whereas the old-task weights remain unchanged.

**Method Overview:** Our method overview is depicted as follows. The CPG method compresses the deep model and (selectively) expands the architecture alternatively. First, a compressed model is built from pruning. Given a new task, the weights of the old-task models are fixed as well. Next, we pick and re-use some of the old-task weights critical to the new task via a differentiable mask, and use the previously released weights for learning together. If the accuracy goal is not attained yet, the architecture can be expanded by adding filters or nodes in the model and resuming the procedure. Then we repeat the gradual pruning [51] (i.e., iteratively removing a portion of weights and retraining) for compacting the model of the new task. An overview of our CPG approach is given in Figure 1. The new-task weights are formed by a combination of two parts: the first part is picked via a learnable mask on the old-task weights, and the second part is learned by gradual pruning/retraining of the extra weights. As the old-task weights are only picked but fixed, we can integrate the required function mappings in a compact model without affecting their accuracy in inference. Main characteristics of our approach are summarized as follows.

**Avoid forgetting**: Our approach ensures unforgetting. The function mappings previously built are maintained as exactly the same when new tasks are incrementally added.

**Expand with shrinking**: Our method allows expansion but keeps the compactness of the architecture, which can potentially handle unlimited sequential tasks. Experimental results reveal that multiple tasks can be condensed in a model with slight or no architecture growing.

**Compact knowledge base**: Experimental results show that the condensed model recorded for previous tasks serves as knowledge base with accumulated experience for weights picking in our approach, which yields performance enhancement for learning new tasks.

## 2   Related Work

Continual lifelong learning [28] can be divided into three main categories: network regularization, memory or data replay, and dynamic architecture. Besides, works on task-free [2] or as a program synthesis [43] have also been studied recently. In the following, we give a brief review of works in the main categories, and readers are suggested to refer to a recent survey paper [28] for more studies.

**Network regularization**: The key idea of network regularization approaches [14, 49, 19, 35, 40, 5, 6] is to restrictively update learned model weights. To keep the learned task information, some penalties are added to the change of weights. EWC [14] uses Fisher's information to evaluate the importance of weights for old tasks, and updates weights according to the degree of importance. Based on similar ideas, the method in [49] calculates the importance by the learning trajectory. Online EWC [40] and EWC++ [5] improve the efficiency issues of EWC. Learning without Memorizing(LwM) [6] presents an information preserving penalty. The approach builds an attention map, and hopes that the attention region of the previous and concurrent models are consistent. These works alleviate catastrophic forgetting but cannot guarantee the previous-task accuracy exactly.

**Memory replay**: Memory or data replay methods [32, 41, 13, 3, 46, 45, 11, 34, 33, 27] use additional models to remember data information. Generative Replay [41] introduces GANs to lifelong learning. It uses a generator to sample fake data which have similar distribution to previous data. New tasks can be trained with these generated data. Memory Replay GANs (MeRGANs) [45] shows that forgetting phenomenon still exists in a generator, and the property of generated data will become worse with incoming tasks. They use replay data to enhance the generator quality. Dynamic Generative Memory (DGM) [27] uses neural masking to learn connection plasticity in conditional generative models, and set a dynamic expansion mechanism in the generator for sequential tasks. Although these methods can exploit data information, they still cannot guarantee the exact performance of past tasks.

**Dynamic architecture**: Dynamic-architecture approaches [38, 20, 36, 29, 48] adapt the architecture with a sequence of tasks. ProgressiveNet [38] expands the architecture for new tasks and keeps the function mappings by preserving the previous weights. LwF [20] divides the model layers into two parts, shared and task-specific, where the former are co-used by tasks and the later are grown with further branches for new tasks. DAN [36] extends the architecture per new task, while each layer in the new-task model is a sparse linear-combination of the original filters in the corresponding layer of a base model. Architecture expansion has also been adopted in a recent memory-replay approach [27] on GANs. These methods can considerably lessen or avoid catastrophic forgetting via architecture expansion, but the model is monotonically increased and a redundant structure would be yielded.

As continually growing the architecture will retain the model redundancy, some approach performs model compression before expansion [48] so that a compact model can be built. In the past, the most related method to ours is Dynamic-expansion Net (DEN) [48]. DEN reduces the weights of the previous tasks via sparse-regularization. Newly added weights and old weights are both adapted for the new task with sparse constraints. However, DEN does not ensure non-forgetting. As the old-task weights are jointly trained with the new weights, part of the old-tasks weights are selected and modified. Hence, a "Split & Duplication" step is introduced to further 'restore' some of the old weights modified for lessening the forgetting effect. Pack and Expand (PAE) [12] is our previous approach that takes advantage of PackNet [23] and ProgressiveNet [38]. It can avoid forgetting, maintain model compactness, and allow dynamic model expansion. However, as it uses all weights of previous tasks for sharing, the performance becomes less favorable when learning a new task.

Our approach (CPG) is accomplished by a compacting→picking(→growing) loop, which selects critical weights from old tasks without modifying them, and thus avoids forgetting. Besides, our approach does not have to restore the old-task performance like DEN as the performance is already kept, which thus avoids a tedious "Split & Duplication" process which takes extra time for model adjustment and will affect the new-task performance. Our approach is hence simple and easier to implement. In the experimental results, we show that our approach also outperforms DEN and PAE.

## 3 The CPG approach for Continual Lifelong Learning

Without loss of generality, our work follows a task-based sequential learning setup that is a common setting in continual learning. In the following, we present our method in the sequential-task manner.

**Task 1**: Given the first task (task-1) and an initial model trained via its dataset, we perform gradual pruning [51] on the model to remove its redundancy with the performance kept. Instead of pruning weights one time to the pruning ratio goal, the gradual pruning removes a portion of the weights and retrains the model to restore the performance iteratively until meeting the pruning criteria. Thus, we compact the current model so that redundancy among the model weights are removed (or released). The weights in the compact model are then set unalterable and remain fixed to avoid forgetting. After

gradual pruning, the model weights can be divided into two parts: one is preserved for task 1; the other is released and able to be employed by the subsequent tasks.

**Task k to k+1**: Assume that in task-$k$, a compact model that can handle tasks 1 to $k$ has been built and available. The model weights preserved for tasks 1 to $k$ are denoted as $\mathbf{W}_{1:k}^P$. The released (redundant) weights associated with task-$k$ are denoted as $\mathbf{W}_k^E$, and they are extra weights that can be used for subsequent tasks. Given the dataset of task-$(k+1)$, we apply a learnable mask $\mathbf{M}$ to pick the old weights $\mathbf{W}_{1:k}^P$, $\mathbf{M} \in \{0,1\}^D$ with $D$ the dimension of $\mathbf{W}_{1:k}^P$. The weights picked are then represented as $\mathbf{M} \odot \mathbf{W}_{1:k}^P$, the element-wise product of the 0-1 mask $\mathbf{M}$ and $\mathbf{W}_{1:k}^P$. Without loss of generality, we use the piggyback approach [22] that learns a real-valued mask $\hat{\mathbf{M}}$ and applies a threshold for binarization to construct $\mathbf{M}$. Hence, given a new task, we pick a set of weights (known as the critical weights) from the compact model via a learnable mask. Besides, we also use the released weights $\mathbf{W}_k^E$ for the new task. The mask $\mathbf{M}$ and the additional weights $\mathbf{W}_k^E$ are learned together on the training data of task-$(k+1)$ with the loss function of task-$(k+1)$ via back-propagation. Since the binarized mask $\mathbf{M}$ is not differentiable, when training the binary mask $\mathbf{M}$, we update the real-valued mask $\hat{\mathbf{M}}$ in the backward pass; then $\mathbf{M}$ is quantized with a threshold on $\hat{\mathbf{M}}$ and applied to the forward pass. If the performance is unsatisfied yet, the model architecture can be grown to include more weights for training. That is, $\mathbf{W}_k^E$ can be augmented with additional weights (such as new filters in convolutional layers and nodes in fully-connected layers) and then resumes the training of both $\mathbf{M}$ and $\mathbf{W}_k^E$. Note that during traning, the mask $\mathbf{M}$ and new weights $\mathbf{W}_k^E$ are adapted but the old weights $\mathbf{W}_{1:k}^P$ are "picked" only and remain fixed. Thus, old tasks can be exactly recalled.

**Compaction of task k+1**: After $\mathbf{M}$ and $\mathbf{W}_k^E$ are learned, an initial model of task-$(k+1)$ is obtained. Then, we fix the mask $\mathbf{M}$ and apply gradual pruning to compress $\mathbf{W}_k^E$, so as to get the compact model $\mathbf{W}_{k+1}^P$ and the redundant (released) weights $\mathbf{W}_{k+1}^E$ for task-$(k+1)$. The compact model of old tasks then becomes $\mathbf{W}_{1:(k+1)}^P = \mathbf{W}_{1:k}^P \cup \mathbf{W}_{k+1}^P$. The compacting and picking/growing loop is repeated from task to task. Details of CPG continual learning is listed in Algorithm 1.

---

**Algorithm 1:** Compacting, Picking and Growing Continual Learning

---
**Input:** given task 1 and an original model trained on task 1.

Set an accuracy goal for task 1;

Alternatively remove small weights and re-train the remaining weights for task 1 via gradual pruning [51], whenever the accuracy goal is still hold;

Let the model weights preserved for task 1 be $\mathbf{W}_1^P$ (referred to as task-1 weights), and those that are removed by the iterative pruning be $\mathbf{W}_1^E$ (referred to as the released weights);

**for** *task $k = 2 \cdots K$ (let the released weights of task $k$ be $W_k^E$)* **do**

    Set an accuracy goal for task $k$;

    Apply a mask $\mathbf{M}$ to the weights $\mathbf{W}_{1:k-1}^P$; train both $\mathbf{M}$ and $\mathbf{W}_{k-1}^E$ for task $k$, with $\mathbf{W}_{1:k-1}^P$ fixed;

    If the accuracy goal is not achieved, expand the number of filters (weights) in the model, reset $\mathbf{W}_{k-1}^E$ and go to previous step;

    Gradually prune $\mathbf{W}_{k-1}^E$ to obtain $\mathbf{W}_k^E$ (with $\mathbf{W}_{1:k-1}^P$ fixed) for task $k$, until meeting the accuracy goal;

    $\mathbf{W}_k^P = \mathbf{W}_{k-1}^E \backslash \mathbf{W}_k^E$ and $\mathbf{W}_{1:k}^P = \mathbf{W}_{1:k-1}^P \cup \mathbf{W}_k^P$;

**end**

---

## 4   Experiments and Results

We perform three experiments to verify the effectiveness of our approach. The first experiment contains 20 tasks organized with CIFAR-100 dataset [16]. In the second experiment, we follow the same settings of PackNet [23] and Piggyback [22] approaches, where several fine-grained datasets are chosen for classification in an incremental manner. In the third experiment, we start from face verification and compact three further facial-informatic tasks (expression, gender, and age) incrementally to examine the performance of our continual learning approach in a realistic scenario. We implement our CPG approach[1] and independent task learning (from scratch or fine-tuning) via PyTorch [30] in all experiments, but implement DEN [27] via Tensorflow [1] with its official codes.

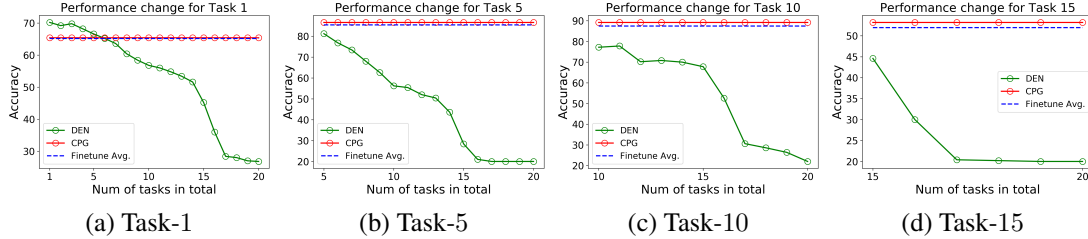

|                (a) Task-1                |                (b) Task-5                |                (c) Task-10                |                (d) Task-15                |

Figure 2: The accuracy of DEN, Finetune and CPG for the sequential tasks 1, 5, 10, 15 on CIFAR-100.

## 4.1 Twenty Tasks of CIFAR-100

We divide the CIFAR-100 dataset into 20 tasks. Each task has 5 classes, 2500 training images, and 500 testing images. In the experiment, VGG16-BN model (VGG16 with batch normalization layers) is employed to train the 20 tasks sequentially. First, we compare our approach with DEN [27] (as it also uses an alternating mechanism of compression and expansion) and fine-tuning. To implement fine-tuning, we train task-1 from scratch by using VGG16-BN; then, assuming the models of task 1 to task $k$ are available, we then train the model of task-$(k + 1)$ by fine-tuning one of the models randomly selected from tasks 1 to $k$. We repeat this process 5 times and get the average accuracy (referred to as Finetune Avg). To implement our CPG approach, task-1 is also trained by using VGG16-BN, and this initial model is adapted for the sequential tasks following Algorithm 1. DEN is implemented via the official codes provided by the authors and modified for VGG16-BN.

Figure 2 shows the classification accuracy of DEN, fine-tuning, and our CPG. Figure 2(a) is the accuracy of task-1 when all of the 20 tasks have been trained. Initially, the accuracy of DEN is higher than CPG and fine-tuning although the same model is trained from scratch. We conjecture that it is because they are implemented on different platforms (Tensorflow vs PyTorch). Nevertheless, the performance of task-1 gradually drops when the other tasks (2 to 20) are increasingly learned in DEN, as shown in Figure 2(a), and the drops are particularly significant for tasks 15 to 20. In Figure 2(b), the initial accuracy of DEN on task-5 becomes a little worse than that of CPG and fine-tuning. It reveals that DEN could not employ the previously leaned model (tasks 1-4) to enhance the performance of the current task (task 5). Besides, the accuracy of task-5 still drops when new tasks (6-20) are learned. Similarly, for tasks 10 and 15 respectively shown in Figures 2(c) and (d), DEN has a large performance gap on the initial model, with an increasing accuracy dropping either.

We attribute the phenomenon as follows. As DEN does not guarantee unforgetting, a "Split & Duplication" step is enforced to recover the old-task performance. Though DEN tries to preserve the learned tasks as much as they could via optimizing weight sparsity, the tuning of hyperparameters in its loss function makes DEN non-intuitive to balance the learning of the current task and remembering the previous tasks. The performance thus drops although we have tried our best for tuning it. On the other hand, fine-tuning and our CPG have roughly the same accuracy initially on task-1 (both are trained from scratch), whereas CPG gradually outperforms fine-tuning on tasks 5, 10, and 15 in Figure 2. The results suggest that our approach can exploit the accumulated knowledge base to enhance the new task performance. After model growing for 20 tasks, the final amount of weights is increased by 1.09 times (compared to VGG16-BN) for both DEN and CPG. Hence, our approach can not only ensure maintaining the old-task performance (as the horizontal lines shown in Figure 2), but effectively accumulate the weights for knowledge picking.

Unlike ProgressiveNet that uses all of the weights kept for the old tasks when training the new task, our method only picks the old-task weights critical to the new tasks. To evaluate the effectiveness of the weights picking mechanism, we compare CPG with PAE [12] and PackNet [23]. In our method, if all of the old weights are always picked, it is referred to as the pack-and-expand (PAE) approach. If we further restrict PAE such that the architecture expansion is forbidden, it degenerates to an existing approach, PackNet [23]. Note that both PAE and PackNet ensure unforgetting. As shown in Table 1, besides the first two tasks, CPG performs more favorably than PAE and PackNet consistently. The results reveal that the critical-weights picking mechanism in CPG not only reduces the unnecessary weights but also boost the performance for new tasks. As PackNet does not allow model expansion, its weights amount remains the same ($1\times$). However, when proceeding with more tasks, available space in PackNet gradually reduces, which limits the effectiveness of PackNet to learn new tasks. PAE uses all the previous weights during learning. As with more tasks, the weights from previous

Table 1: The performance of PackNet, PAE and CPG on CIFAR-100 twenty tasks. We use Avg., Exp. and Red. as abbreviations for Average accuracy, Expansion weights and Redundant weights.

| Methods | 1 | 2 | 3 | 4 | 5 | 6 | 7 | 8 | 9 | 10 | 11 | 12 | 13 | 14 | 15 | 16 | 17 | 18 | 19 | 20 | Avg. | Exp. (×) | Red. (×) |
|---|---|---|---|---|---|---|---|---|---|---|---|---|---|---|---|---|---|---|---|---|---|---|---|
| **PackNet** | 66.4 | 80.0 | 76.2 | 78.4 | 80.0 | 79.8 | 67.8 | 61.4 | 68.8 | 77.2 | 79.0 | 59.4 | 66.4 | 57.2 | 36.0 | 54.2 | 51.6 | 58.8 | 67.8 | 83.2 | 67.5 | 1 | 0 |
| **PAE** | 67.2 | 77.0 | 78.6 | 76.0 | 84.4 | 81.2 | 77.6 | 80.0 | 80.4 | 87.8 | 85.4 | 77.8 | 79.4 | 79.6 | 51.2 | 68.4 | 68.6 | 68.6 | 83.2 | 88.8 | 77.1 | 2 | 0 |
| **CPG** | 65.2 | 76.6 | 79.8 | 81.4 | 86.6 | 84.8 | 83.4 | 85.0 | 87.2 | 89.2 | 90.8 | 82.4 | 85.6 | 85.2 | 53.2 | 74.4 | 70.0 | 73.4 | 88.8 | 94.8 | 80.9 | 1.5 | 0.41 |

Table 2: The performance of CPGs and individual models on CIFAR-100 twenty tasks. We use fine-Avg and fine-Max as abbreviations for Average and Max accuracy of the 5 fine-tuning models.

| Methods | 1 | 2 | 3 | 4 | 5 | 6 | 7 | 8 | 9 | 10 | 11 | 12 | 13 | 14 | 15 | 16 | 17 | 18 | 19 | 20 | Avg. | Exp. (×) | Red. (×) |
|---|---|---|---|---|---|---|---|---|---|---|---|---|---|---|---|---|---|---|---|---|---|---|---|
| **Scratch** | 65.8 | 78.4 | 76.6 | 82.4 | 82.2 | 84.6 | 78.6 | 84.8 | 83.4 | 89.4 | 87.8 | 80.2 | 84.4 | 80.2 | 52.0 | 69.4 | 66.4 | 70.0 | 87.2 | 91.2 | 78.8 | 20 | 0 |
| **fine-Avg** | 65.2 | 76.1 | 76.1 | 77.8 | 85.4 | 82.5 | 79.4 | 82.4 | 82.0 | 87.4 | 87.4 | 81.5 | 84.6 | 80.8 | 52.0 | 72.1 | 68.1 | 71.9 | 88.1 | 91.5 | 78.6 | 20 | 0 |
| **fine-Max** | 65.8 | 76.8 | 78.6 | 80.0 | 86.2 | 84.8 | 80.4 | 84.0 | 83.8 | 88.4 | 89.4 | 83.8 | 87.2 | 82.8 | 53.6 | 74.6 | 68.8 | 74.4 | 89.2 | 92.2 | 80.2 | 20 | 0 |
| **CPG avg** | 65.2 | 76.6 | 79.8 | 81.4 | 86.6 | 84.8 | 83.4 | 85.0 | 87.2 | 89.2 | 90.8 | 82.4 | 85.6 | 85.2 | 53.2 | 74.4 | 70.0 | 73.4 | 88.8 | 94.8 | 80.9 | 1.5 | 0.41 |
| **CPG max** | 67.0 | 79.2 | 77.2 | 82.0 | 86.8 | 87.2 | 82.0 | 85.6 | 86.4 | 89.6 | 90.0 | 84.0 | 87.2 | 84.8 | 55.4 | 73.8 | 72.0 | 71.6 | 89.6 | 92.8 | 81.2 | 1.5 | 0 |
| **CPG top** | 66.6 | 77.2 | 78.6 | 83.2 | 88.2 | 85.8 | 82.4 | 85.4 | 87.6 | 90.8 | 91.0 | 84.6 | 89.2 | 83.0 | 56.2 | 75.4 | 71.0 | 73.8 | 90.6 | 93.6 | 81.7 | 1.5 | 0 |

tasks would dominate the whole network and become a burden in learning new tasks. Finally, as shown in the Expand (Exp.) field in Table 1, PAE grows the model and uses 2 times of weights for the 20 tasks. Our CPG expands to $1.5\times$ of weights (with $0.41\times$ redundant weights that can be released to future tasks). Hence, CPG finds a more compact and sustainable model with better accuracy when the picking mechanism is enforced.

Table 2 shows the performance of different settings of our CPG method, together with their comparison to independent task learning (including learning from scratch and fine-tuning from a pre-trained model). In this table, 'scratch' means learning each task independently from scratch via the VGG16-BN model. As depicted before, 'fine-Avg' means the average accuracy of fine-tuning from a previous model randomly selected and repeats the process 5 times. 'fine-Max.' means the maximum accuracy of these 5 random trials. In the implementation of our CPG algorithm, an accuracy goal has to be set for the gradual-pruning and model-expansion steps. In this table, the 'avg', 'max', and 'top' correspond to the settings of accuracy goals to be fine-Avg, fine-Max, and a slight increment of the maximum of both, respectively. The upper bound of model weights expansion is set as 1.5 in this experiment. As can be seen in Table 2, CPG gets better accuracy than both the average and maximum of fine-tuning in general. CPG also performs more favorably than learning from scratch averagely. This reveals again that the knowledge previously learned with our CPG can help learn new tasks.

Besides, the results show that a higher accuracy goal yields better performance and more consumption of weights in general. In Table 2, the accuracy achieved by 'CPG avg', 'CPG max', and 'CPG top' is getting increased. The former remains to have $0.41\times$ redundant weights that are saved for future use, whereas the later two consume all weights. The model size includes not only the backbone model weights, but also the overhead of final layers increased with new classes, batch-normalization parameters, and the binary masks. Including all overheads, the model sizes of CPG for the three settings are $2.16\times$, $2.40\times$ and $2.41\times$ of the original VGG16-BN, as shown in Table 3. Compared to independent models (learning-from-scratch or fine-tuning) that require $20\times$ for maintaining the old-task accuracy, our approach can yield a far smaller model to achieve exact unforgetting.

## 4.2 Fine-grained Image Classification Tasks

In this experiment, following the same settings in the works of PackNet [23] and Piggyback [22], six image classification datasets are used. The statistics are summarized in Table 4, where ImageNet [17] is the first task, following by fine-grained classification tasks, CUBS [44], Stanford Cars [15] and Flowers [26], and finally WikiArt [39] and Sketch [8] that are artificial images drawing in various styles and objects. Unlike previous experiments where the first task consists of some of the five classes from CIFAR-100. In this experiment, the first-task classifier is trained on ImageNet, which is a strong base for fine-tuning. Hence, in the fine-tuning setting of this experiment, tasks 2 to 6 are all fine-tuned from the task-1, instead of selecting a previous task randomly. For all tasks, the image size is $224 \times 224$, and the architecture used in this experiment is ResNet50.

Table 3: Model sizes on CIFAR-100 twenty tasks.

| Methods | Model Size (MB) |
|---|---|
| VGG16-BN | 128.25 |
| Individual Models | 2565 |
| CPG avg | 278 |
| CPG max | 308 |
| CPG top | 310 |

Table 4: Statistics of the fine-grained datasets

| Dataset | #Train | #Eval | #Classes |
|---|---|---|---|
| ImageNet | 1,281,167 | 50,000 | 1,000 |
| CUBS | 5,994 | 5,794 | 200 |
| Stanford Cars | 8,144 | 8,041 | 196 |
| Flowers | 2,040 | 6,149 | 102 |
| WikiArt | 42,129 | 10,628 | 195 |
| Sketch | 16,000 | 4,000 | 250 |

Table 5: Statistics of the facial-informatic datasets

| Dataset | #Train | #Eval | #Classes |
|---|---|---|---|
| VGGFace2 | 3,137,807 | 0 | 8,6301 |
| LFW | 0 | 13,233 | 5,749 |
| FotW | 6,171 | 3,086 | 3 |
| IMDB-Wiki | 216,161 | 0 | 3 |
| AffectNet | 283,901 | 3,500 | 7 |
| Adience | 12,287 | 3,868 | 8 |

Table 6: Accuracy on fine-grained dataset.

| Dataset | Train from Scratch | Finetune | Prog. Net | PackNet | Piggyback | CPG |
|---|---|---|---|---|---|---|
| ImageNet | 76.16 | - | 76.16 | 75.71 | 76.16 | 75.81 |
| CUBS | 40.96 | 82.83 | 78.94 | 80.41 | 81.59 | 83.59 |
| Stanford Cars | 61.56 | 91.83 | 89.21 | 86.11 | 89.62 | 92.80 |
| Flowers | 59.73 | 96.56 | 93.41 | 93.04 | 94.77 | 96.62 |
| Wikiart | 56.50 | 75.60 | 74.94 | 69.40 | 71.33 | 77.15 |
| Sketch | 75.40 | 80.78 | 76.35 | 76.17 | 79.91 | 80.33 |
| Model Size (MB) | 554 | 554 | 563 | 115 | 121 | 121 |

Table 7: Accuracy on facial-informatic tasks.

| Task | Train from Scratch | Finetune | CPG |
|---|---|---|---|
| Face | $99.417 \pm 0.367$ | - | $99.300 \pm 0.384$ |
| Gender | 83.70 | 90.80 | 89.66 |
| Expression | 57.64 | 62.54 | 63.57 |
| Age | 46.14 | 57.27 | 57.66 |
| Exp. ($\times$) | 4 | 4 | 1 |
| Red. ($\times$) | 0 | 0 | 0.003 |

The performance is shown on Table 6. Five methods are compared with CPG: training from scratch, fine-tuning, ProgressiveNet, PackNet, and Piggyback. For the first task (ImageNet), CPG and PackNet performs slightly worse than the others, since both methods have to compress the model (ResNet50) via pruning. Then, for tasks 2 to 6, CPG outperforms the others in almost all cases, which shows the superiority of our method on building a compact and unforgetting base for continual learning. As for the model size, ProgressiveNet increases the model per task. Learning-from-scratch and fine-tuning need 6 models to achieve unforgetting. Their model sizes are thus large. CPG yields a smaller model size comparable to piggyback, which is favorable when considering both accuracy and model size.

## 4.3 Facial-informatic Tasks

In a realistic scenario, four facial-informatic tasks, face verification, gender, expression and age classification are used with the datasets summarized in Table 5. For face verification, we use VGGFace2 [4] for training and LFW [18] for testing. For gender classification, we combine FotW [9] and IMDB-Wiki [37] datasets and classify faces into three categories, male, female and other. The AffectNet dataset [25] is used for expression classification that classifies faces into seven primary emotions. Finally, Adience dataset [7] contains faces with labels of eight different age groups. For these datasets, faces are aligned using MTCNN [50] with output size of $112 \times 112$. We use the 20-layer CNN in SphereFace [21] and train a model for the face verification task accordingly. We compare CPG with the models fine-tuned from the face verification task. The results are reported in Table 7. Compared with individual models (training-from-scratch and fine-tuning), CPG can achieve comparable or more favorable results without additional expansion. After learning four tasks, CPG still has $0.003 \times$ of the released weights able to be used for new tasks.

## 5 Conclusion and Future Work

We introduce a simple but effective method, CPG, for continual learning avoiding forgetting. Compacting a model can prevent the model complexity from unaffordable when the number of tasks is increased. Picking learned weights using binary masks and train them together with newly added weights is an effective way to reuse previous knowledge. The weights for old tasks are preserved, and thus prevents them from forgetting. Growing the model for new tasks facilitates the model to learn unlimited and unknown/un-related tasks. CPG is easy to be realized and applicable to real situations. Experiments show that CPG can achieve similar or better accuracy with limited additional space. Currently, our method compacts a model by weights pruning, and we plan to include channel pruning in the future. Besides, we assume that clear boundaries exit between the tasks, and will extend our approach to handle continual learning problems without task boundaries. In the future, We also plan to provide a mechanism of "selectively forgetting" some previous tasks via the masks recorded.

**Acknowledgments**

We thank the anonymous reviewers and area chair for their constructive comments. This work is supported in part under contract MOST 108-2634-F-001-004.

## Footnotes

[1]Our codes are available at https://github.com/ivclab/CPG.

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
