[Supplementary Material]

# Supplementary Material of "Compacting, Picking and Growing for Unforgetting Continual Learning," NeurIPS 2019

In this supplementary material, we report additional experimental results on CIFAR-100 dataset. Remember that we have divided this dataset into 20 tasks. Each task has 5 classes.

**Random Order of Tasks**: We show that our approach remains effective when the order of tasks is changed. To verify this, we reshuffle the 20 tasks and apply our continual-learning method to the tasks of new orders. The results are shown in Table 8. In this table, CFP-ord.$i$, ($i = 1 \cdots 3$) are the three re-ordering sequences of tasks, and CFP-ord.0 is the case of original ordering (with the same results of CPG shown in Table 1). As can be seen, the average accuracy of the 20 tasks is roughly the same for the random-ordering settings, which reveals that our approach is potentially applicable to continual lifelong learning when the order of tasks is arbitrary.

Table 8: The performance of CPG on CIFAR-100 twenty tasks in random order.

| Methods | 1 | 2 | 3 | 4 | 5 | 6 | 7 | 8 | 9 | 10 | 11 | 12 | 13 | 14 | 15 | 16 | 17 | 18 | 19 | 20 | Avg. | Exp. ($\times$) | Red. ($\times$) |
|---|---|---|---|---|---|---|---|---|---|---|---|---|---|---|---|---|---|---|---|---|---|---|---|
| **CPG-ord.0** | 65.2 | 76.6 | 79.8 | 81.4 | 86.6 | 84.8 | 83.4 | 85.0 | 87.2 | 89.2 | 90.8 | 82.4 | 85.6 | 85.2 | 53.2 | 74.4 | 70.0 | 73.4 | 88.8 | 94.8 | 80.9 | 1.5 | 0.41 |

| Methods | 12 | 9 | 3 | 2 | 7 | 15 | 1 | 19 | 13 | 16 | 8 | 14 | 10 | 20 | 5 | 4 | 18 | 17 | 6 | 11 | Avg. | Exp. ($\times$) | Red. ($\times$) |
|---|---|---|---|---|---|---|---|---|---|---|---|---|---|---|---|---|---|---|---|---|---|---|---|
| **CPG-ord.1** | 81.0 | 83.8 | 78.6 | 79.4 | 82.4 | 52.4 | 67.4 | 89.6 | 86.0 | 74.8 | 85.0 | 85.8 | 87.6 | 92.4 | 87.2 | 80.0 | 72.6 | 71.2 | 86.0 | 89.8 | 80.7 | 1.30 | 0 |

| Methods | 2 | 11 | 1 | 17 | 5 | 4 | 10 | 3 | 9 | 13 | 7 | 15 | 6 | 14 | 12 | 18 | 19 | 16 | 8 | 20 | Avg. | Exp. ($\times$) | Red. ($\times$) |
|---|---|---|---|---|---|---|---|---|---|---|---|---|---|---|---|---|---|---|---|---|---|---|---|
| **CPG-ord.2** | 73.0 | 88.2 | 65.8 | 70.0 | 87.6 | 80.6 | 88.4 | 79.2 | 85.0 | 87.2 | 80.6 | 52.2 | 86.2 | 84.0 | 83.0 | 73.2 | 90.2 | 74.4 | 83.0 | 94.2 | 80.3 | 1.5 | 0 |

| Methods | 19 | 13 | 18 | 10 | 17 | 15 | 6 | 14 | 11 | 16 | 20 | 5 | 9 | 2 | 4 | 8 | 7 | 12 | 1 | 3 | Avg. | Exp. ($\times$) | Red. ($\times$) |
|---|---|---|---|---|---|---|---|---|---|---|---|---|---|---|---|---|---|---|---|---|---|---|---|
| **CPG-ord.3** | 86.0 | 87.8 | 70.0 | 89.0 | 69.6 | 53.6 | 85.2 | 84.8 | 88.6 | 75.6 | 92.4 | 87.2 | 87.0 | 78.0 | 79.6 | 84.0 | 81.8 | 83.2 | 66.4 | 79.2 | 80.5 | 1.4 | 0 |