[Reviews · NeurIPS 2019]

Reviewer 1



The proposed approach is compelling in its simplicity -- this is a good thing, especially given its effectiveness. It is more of an engineering solution than a mathematically motivated method. It is highly effective, and presented clearly. The individual aspects of the approach (compaction, retention of critical connections, etc.) have been explored in individual methods, but this seems a compelling and novel combination of these methods. I liked the ablative dissection of the approach into existing methods as special cases in Section 1: Method Overview. As mentioned, this approach is more engineering than mathematically principled. That's fine given its effectiveness, but it would be useful to develop or combine the mathematical foundations to better motivate the approach. The use of the retention masks would seem to prevent new tasks from fine-tuning previously learned models, limiting the amount of reverse transfer. Progressive nets is incapable of reverse transfer, which is another large limitation of that approach. Is your CPG approach similarly limited in this way, and if not, how is it capable of fine-tuning previous models based on new tasks? The experimental analysis is good, but lacks key details on the specific setups for all experiments. These should have been added as supplemental material, and the final paper would need to be revised to include them. Tables 1 and 2 would be much better presented as a line or bar graph. Are these peak performances or performances after training all tasks (which might be the same)? If the method did incur some sort of reverse transfer or fine-tuning from subsequent learning, it would be good to examine learning curves of all tasks over time, and also to measure any amount of forgetting. POST-RESPONSE Thanks for your comments. The biological motivation you mentioned isn't near as satisfactory as a solid mathematical motivation, although it would be quite difficult to come up with the math/theory behind such an engineered approach.

Reviewer 2



Summary: This paper uses three already existing approaches in continual learning, PackNet, Piggyback, and ProgressiveNets to reduce forgetting. Upon learning a task, as in PackNet and Ref#49, the network is gradually pruned, then Piggyback is used to learn a differentiable mask, and when network’s capacity is maxed out, the architecture can be expanded similar to ProgressiveNets. While the individual parts are not novel, combining all three of them in one model is still interesting. Strength: a) Overall, the paper is well-written and organized. b) The approach is overall novel and shows how to combine three existing ideas (PackNet, Piggyback, and ProgressiveNet) c) Experimental evaluation on several larger scale datasets as tasks show that the model outperforms prior work w.r.t. accuracy at a small model size footprint. d) Additional evaluation is also shown on CIFAR (less convincing) and different Face recognition tasks Weaknesses: 1. Determining hyperparameters and reporting complexity 1.1. The paper requires setting “accuracy goals” when encountering a new task. However, it might be unclear which accuracy can be reached and the paper is opaque how these accuracy goals are determined e.g. when comparing to prior work. To reach optimal performance algorithm 1 might need significant manual intervention. 1.1.1. How are the “accuracy goals” determined (especially for Table 6,7)? 1.1.2. What happens if growing the network does not lead to achieving the accuracy goal? E.g. increasing the network capacity might lead to stronger overfitting and a reduced accuracy? 1.2. The approach may need many iterations to retrain the model to meet the “accuracy goal” (both w.r.t. growing and compressing) 1.3. How much is the model grown, how much is picked, how much is compressed? It would be interesting to see this for the different models in Table 6, as well as the accuracy targets. 1.4. It would be good to report the memory overhead from the binary masks and relate this to memory-based approached such as GEM, A-GEM, and generative replay. 2. Experimental Evaluation 2.1. Ablations 2.1.1. The paper claims that “Another distinction of our approach is the “picking” step “. However, this aspect is not ablated. 2.2. Experiments on CIFAR. The comparison on CIFAR is not convincing 2.2.1. The continual learning literature has extensive experiments on this dataset and the paper only compares to one approach (DEN). 2.2.2. It is unclear if DEN is correctly used/evaluated. It would have been more convincing if the authors used the same setup as in the DEN paper to make sure the comparison is fair/correct. 3. Motivation 3.1. The paper claims forgetting is fully avoided due to the usage of a mask. While it is true that *after* model compression no further forgetting happens, but there is an accuracy drop during pruning, in contrast to e.g. regularization-based methods. Specifically, the original value (before pruning) is not recoverable and hence should be reported as forgetting. 4. The checklist is not fully accurate. The paper does not provide error bars and std-deviation for experiments. 5. Minor: 5.1. Grammar issue in word “determining” in the 4th paragraph on page 3. 5.2. On page 3, in “Method overview” it says “An overview of our method is depicted below” whereas it should directly refer to Figure 1 because Figure 1 is on page 2 5.3. On page 6, right below Figure 2, it says “in all experiments, but realize DEN”. Word “realize” does not fit into the context. 5.4. In future, please use the submission template (not the camera-ready version) so that line numbers on the margins can be used to easily refer to the text. I lean more towards accept: The overall convincing results (especially Table 6) and overall novel model outweigh the limitations discussed above.

Reviewer 3



I found this sentence convincing "replay needs re-training which requires memory" Iterative pruning is a big overhead after learning each task The authors stated that "Without loss of generality, our work follows a task-based sequential learning setup" It is a standard setup but it has many limitations, not being applied when tasks are not known at test time for example. How could this not limit the generality of the approach? Why gradual pruning, there are many methods for compression. Choice is not discussed extensively. The main components of the methods are: 1- Compression, 2- piggyback(a previous method) and training of released weight. 3-If not enough (no explanation of what this means), add nodes or filter. How is this happening??? 4- The algorithm is a paragraph of text. 5-Comparison to HAT is missing, an icml18 paper that masks neurons instead of parameters. "Overcoming Catastrophic Forgetting with Hard Attention to the Task" No explanation on how to behave at test time. Are weights only associated with the test task activated? How does this limit the method?

[Author Response · NeurIPS 2019]

R1: Thank you for recognizing this method as simple and novel (though combining existing concepts). We will provide line or bar graphs in the experiments and add more details on the specific setups in supplementary materials. **[Reverse transfer]** The issue you raised about reverse-transferring to previous models is important and interesting. Existing method (like DEN[46]) solves it by sharing and fine-tuning partial old-task weights with the new task. However, lacking the old training data causes the accuracy degradation of old tasks in subsequent learning. We plan to integrate memory-replay principle to address the reverse-transfer issue. Current replaying methods suffer from re-training which requires memory and choosing an architecture suitable for all tasks. Our method could address these issues by recording partially key data complemented to the preserved weights for fine-tuning old-task models. Finding complemented data for reserve-transfer would be an interesting topic and will be discussed as a future work of our study. **[Math. motivation]** Our method is not motivated by mathematics actually. Instead, it is more bio-motivated. Compacting the model can be seen as a 'consolidation' step in our brain; consolidation of recent memories into long-term storage occurs during REM sleep (Gais et al. 2007). Growing the model by increasing the neurons corresponds to 'neurogenesis' in the human brain. Adult neurogenesis contributes to the formation of new memories (Eriksson et al. 1998, Gage 2000). Hippocampal neurogenesis sustains human-specific cognitive function throughout life (Boldrini et al. 2018). Picking mechanism results in different cognitive functions relying on canonical neural circuits replicated across multiple brain areas (Douglas et al. 1995). Thus, compacting, picking and growing modules are effectively and sustainably combined.

R2: Thank you for leaning more towards accept. **1.1** The accuracy goals are set based on that obtained by fine-tuning the current task from an old task model. So, in Table 6, the goal of task $k$ ($k = 2 \cdots 6$) is set as the accuracy of fine-tuning the ImageNet model (task 1) to task $k$. On Table 7, the goals are determined by fine-tuning the FaceRec model of task 1. We set a maximum size for growing; While not meeting the accuracy goal, the CPG process will stop when running out the maximum size. **1.2** Assume the iteration time is $T$, our gradual pruning trains the 10% weights-pruned model with $0.1T$ iterations, then 20% weights-pruned model with a further $0.1T$ iterations, and so on. The total iterations $T$ is set the same as a common pruning procedure, which has not to be particularly large. **1.3** The "Finetune" column is the accuracy target. In this setting, the CPG model achieves six tasks with the only size of weights equal to one model. It is compressed to 0.6 in the first task, then expand to 0.62, 0.81, 0.8195, 0.98195, and 1.The pickup ratios from the first to fifth tasks are 0.756, 0.765, 0.822, 0.581, and 0.667. If we allow the model to double, the accuracy will increase a little. **1.4** We randomly split tasks on CIFAR and use reduced ResNet18 as in A-GEM and obtain average accuracy of 63.4 at the end of the 17-th task, better than A-GEM with 62.2, GEM with 61.2, and Prog-Net with 59.1. The memory overhead of CPG is the binary masks, which costs 2.59 MB, and that of A-GEM is the episodic memory, which costs 3.80 MB. **2.1** A distinction of our approach is the "picking" step. This has been ablated in Exp.-CIFAR, where PAE (using all the previous weights during learning) is a special case of our CPG; As seen, PAE (without picking but using all) performs worse on both model size and accuracy. Our new results also show that PAE is less favorable on larger-scale data. **2.2** We compare our method with most related works, including (1) the methods using also an iterating mechanism of compression and growing, and (2) the methods able to avoid forgetting. As for (1), to our best knowledge, DEN is a representative one among the few. There are merely few works achieving (2) too, where PackNet, Piggback, and ProgressiveNet are compared in our exps. To enhance the CIFAR experiments, we also conduct additional analysis using the CIFAR setting in A-GEM and obtain average accuracy of 70.6, 68.4, 68.6, 69.9, 67.6, 63.4, 65.2, 73.1, 62.4, 68.6, 72.8, 67.4, 65.8, 70.6, 66.6, 67.1, 63.4 at the end of each task. The average accuracy at the later task shows the forgetting measurements of CPG method. To make a fair comparison on the same architecture, we have carefully implemented DEN since its public codes only support feed-forward networks, whereas convolution and other layers in a typical CNN are missing. In our experience, DEN does not avoid unforgetting and thus a "Split&Duplication" step is enforced to lessen forgetting. However, this step is difficult to tune for balancing the current and old tasks, causing its performance degradation. **3** We would like to claim that model compression does not necessarily degrade the accuracy. E.g., in Exp. Fine-grained, the accuracy before pruning is 76.1, 83.3, 92.7, 96.7, 77.1, 80.3, and after gradual pruning is 75.8, 83.5, 92.8, 96.6, 77.1, 80.3, for the 6 tasks respectively. Thus there are only insignificant accuracy drop during pruning and sometimes the accuracy increases. Our recorded model is the compressed one for an old task and 'unforgetting' is defined in terms of the recorded models. **4, 5** Will be refined accordingly.

R3: We thank your comments. Though you fell novelty of the approach is limited, we would like to emphasize that a mix of other approaches is not necessarily un-novel. We have done critical analyses to the pros and cons of the combination, e.g., ablating our method with ProgressiveNet (without compression), PAE (omit picking), PAC (omit picking and growing), and Piggyback (with picking only); our method can achieve unforgetting, model compactness, sustainability, and high accuracy via an effective way to reuse previous knowledge; this is the first continual lifelong method attaining these goals simultaneously to our knowledge. Our work follows a common sequential task-based setup, and we plan to extend it to the case without task boundaries in the future. We choose gradual pruning because we have tried $l_1$-regularizations but found more iterations are needed to converge (this coincides with the survey of Cheng et al. 2018). HAT is conceptually similar to PackNet but compressing in neuron or filter level with a different mechanism; we will cite it. Our current implementation loads the entire model and the mask for each task, yet our method allows loading only the filters associated with the masked weights to speed up the individual task test.

[Meta-Review · NeurIPS 2019]

The submission originally received mixed reviews, putting making it a borderline case. The reviewers appreciated the (relative) simplicity of the approach and the experimental evaluation of large-scale tasks. They also raised concerns about the originality of the approach, though, because all relevant components had already appeared in previous work, and they criticized a lack of mathematical derivations. After the author response, the reviewers discussed the work in detail. Ultimately, the decision was that the submission provides a valuable addition to the field by describing a well-engineered solution. Therefore, the work should be accepted. The authors are strongly encourage to make every possible effort to let other researchers benefit from their work, releasing their code and data, ideally pretrained models etc.